# Amelioration Strategies for Silver Diamine Fluoride: Moving from Black to White

**DOI:** 10.3390/antibiotics12020298

**Published:** 2023-02-02

**Authors:** Amjad Almuqrin, Inder Preet Kaur, Laurence J. Walsh, Chaminda Jayampath Seneviratne, Sobia Zafar

**Affiliations:** School of Dentistry, The University of Queensland, Brisbane, QLD 4006, Australia

**Keywords:** early childhood caries, ECC, silver diamine fluoride, SDF, nanotechnology, nanoparticles, selenium

## Abstract

Topical cariostatic agents have become a reasonable alternative for managing dental caries in young children. Silver diamine fluoride (SDF) is a practical topical approach to arrest caries and avoid extensive and risky dental treatment. However, the literature demonstrates a parental hesitation towards accepting SDF because of black unaesthetic tooth discolouration following application. The rapid oxidation of ionic silver darkens demineralised tooth structure permanently. In this regard, nano-metallic antimicrobials could augment or substitute for silver, and thereby enhance SDF aesthetic performance. Recently, biomedical research has drawn attention to selenium nanoparticles (SeNPs) due to their antimicrobial, antioxidant, and antiviral potencies. Various in vitro studies have examined the effect of SeNPs on the virulence of bacteria. This narrative review explores practical issues when using SDF and suggests future directions to develop it, focusing on antimicrobial metals. Several methods are described that could be followed to reduce the discolouration concern, including the use of nanoparticles of silver, of silver fluoride, or of selenium or other metals with antimicrobial actions. There could also be value in using remineralising agents other than fluoride, such as NPs of hydroxyapatite. There could be variations made to formulations in order to lower the levels of silver and fluoride in the SDF or even to replace one or both of the silver and fluoride components completely. Moreover, since oxidation processes appear central to the chemistry of the staining, adding SeNPs which have antioxidant actions could have an anti-staining benefit; SeNPs could be used for their antimicrobial actions as well. Future research should address the topic of selenium chemistry to optimise how SeNPs would be used with or in place of ionic silver. Incorporating other antimicrobial metals as nanoparticles should also be explored, taking into account the optimal physicochemical parameters for each of these.

## 1. Introduction

Globally, one of the most widespread childhood illnesses is Early Childhood Caries (ECC). Statistically, this represents some 48% of chronic childhood diseases [1], and is the major reason for preventable hospital admissions in children. Given that it affects children who are too young to cooperate with conventional dental treatment, ECC is typically treated invasively under general anaesthesia (GA). From an economic point of view, untreated ECC is a significant burden to the healthcare system. According to a recent study, a single dental GA session costs over USD 1000 (AUD 1793.23 with SD 803.45) [2], and there is often a long GA waitlist for public sector dental treatment. Private treatment under GA is not affordable for many parents. Hence, there is great value in alternative management options to treat and arrest ECC in young pre-cooperative children that can be quick, simple, painless, affordable, effective, and easy to apply.

In recent years, silver diamine fluoride (SDF) has emerged as a popular topical treatment for achieving caries arrest in young children, thereby reducing the need for invasive and costly restorative or surgical dental treatment [1]. SDF is composed of silver, ammonia, fluoride, and water. Silver ions inhibit bacterial growth by reacting with the bacterial cell wall and with intracellular contents, causing reproductive and metabolic disturbances [2,3]. Fluoride at high concentrations also exerts antibacterial actions, and it remineralises tooth structure. Ammonia elevates the pH and acts as a stabiliser [3].

Studies have shown that when SDF at a concentration of 38% is applied semi-annually, an arrest of 81% of carious lesions in the dentine will predictably occur [4,5]. Furthermore, application does not lead to acute complications such as systemic diseases or toxicity [6]. However, there is still practitioner and parental reluctance towards the use of SDF for caries arrest in children [7,8]. The most common reason cited for avoidance is black discolouration of carious lesions following its application, which is unaesthetic [6]. In a systematic review, parents’ decision whether to accept or reject SDF treatment was linked to tooth position; and a low acceptance rate was reported if SDF treatment was proposed for anterior teeth [9]. Dental aesthetic issues are unlikely to have effects on preschool child social interactions or self-esteem [10,11]. Despite this, parental acceptance of SDF is low because of concerns regarding the black appearance of treated sites on the teeth [8,12]. A survey of 920 school children revealed that, unlike their preschool counterparts, those with abnormalities in tooth shape or colour were more likely to experience bullying, with its attendant psychological and emotional impacts [13].

SDF discolouration is caused by oxidation of ionic silver to metallic silver and silver oxide, with subsequent precipitation of silver–protein and silver phosphate complexes on tooth structure. Given this issue, there is a need to find additional antimicrobial agents to augment or replace silver, reducing the need for high concentrations of ionic silver. Any such replacements should reduce the tendency to darken carious lesions. Nanomaterial-based antimicrobial agents have become more widely used for a range of medical applications. Nanoparticles have distinctive characteristics such as a large surface area and enhanced reactivity [14]. In dentistry, nanoparticles have been loaded into various dental materials to provide antibacterial actions [15]. The antimicrobial potential of metallic nanoparticles can be controlled by altering various physio-chemical parameters, such as particle size, shape, and zeta potential, and by altering the method of synthesis or by applying capping agents [16]. Each physio-chemical property is potentially important when evaluating the biological behaviour and impacts of nanoparticles. For instance, nano morphology plays an important role in the fate and performance of nanoparticles. Different nanoparticle shapes give different diffusion rates. A further point is that shape affects steric hindrance when nanoparticles collide with and interact intimately with surfaces [17]. An additional example of nanoscale parameters influencing behaviour is agglomeration, where many nanoparticles come together into clusters due to attraction forces. Agglomerations cause large clusters to form; these have a smaller surface area and so are not as biologically active as separate individual nanoparticles. A range of important nano specifications are illustrated in Figure 1.

Lately, selenium nanoparticles (SeNPs) have been attracting interest in biomedical research owing to their suitable biocompatibility as well as their antibacterial, antifungal, antiviral, and antioxidant effects [18]. SeNPs have been shown to exert antimicrobial actions against multiple species of bacteria [19]. The antibacterial and antioxidant effects of SeNPs in topical anticaries agents has not been investigated thoroughly [19]. The specific aims of this review are to (1) explore how the chemical composition of SDF could be altered to reduce staining and (2) explore the possibility of using alternative nanomaterial-based antimicrobial agents to augment or replace ionic silver, thus lowering the potential to cause black discolouration of teeth.

## 2. Early Childhood Caries: The Ripple Effect

ECC is one of the most common chronic disorders in the world [20]. ECC is defined as “the presence of one or more decayed (non-cavitated or cavitated lesions), missing (due to caries), or filled tooth surfaces in any primary tooth in a child under the age of six” [21]. It is challenging to provide restorative treatment for younger children because of their immaturity as well as their inability and unwillingness to cooperate for treatment by dentists and therapists provided in the dental chair. Recently, the COVID-19 pandemic crisis has worsened the situation regarding caries in children because parents have postponed their children’s dental appointments [22]. Moreover, in communities that are facing major public health issues because of social disadvantage, high rates of caries in the primary dentition may not be a high priority compared with other health needs [23,24].

ECC affects a child’s life in multiple ways, causing pain, infection, nutrition difficulties, interrupted sleep, self-esteem concerns, and aesthetic problems [25,26]. Advanced cases of ECC can cause further complications such as impaired dietary intake, school absence, impaired growth and development, and repeated emergency admissions and hospitalisations for severe dental infections. Due to the cognitive and communication problems of treating preschool and very young children, dental treatment under GA is often needed. In Australia, there is a growing need for paediatric dental GA, and the waitlist in some public healthcare facilities may extend up to two years [27]. Children who undergo dental GA for the treatment of ECC have a reported relapse rate of up to 79%, as they continue to develop new carious lesions and symptoms requiring repeated treatment [28]. Dental GA sessions are a major financial burden for the health care system. According to the National Independent Hospital Pricing Authority, the average direct cost for a typical ECC treatment episode (including extractions and restorations) under GA was AUD 3029 in 2012–2013. Additionally, a retrospective study (from 2018–2019) reported that dental extraction under GA was the most frequent treatment choice when young children presented to emergency departments with severe dental infections [29].

Dental caries is a complex disease that involves multiple microbes (both bacteria and fungi) which emerge due to ecological changes in the dental plaque biofilm. This dysbiosis is driven by diet and other lifestyle factors [30]. There is an imbalance between acidogenic and aciduric microbes on one hand, and health-associated commensal bacteria on the other [31]. Even though dental caries is polymicrobial in nature, *Streptococcus mutans* (*S. mutans*) serves as a key pathogen by initiating the formation of a dense extracellular polymer matrix composed of glucan-rich exopolysaccharides. *S. mutans* and *Streptococcus sobrinus* (*S. sobrinus*) can both produce large quantities of these insoluble glucans [32], which give the biofilm enhanced bulk and adhesiveness and serve as a source of fermentable carbohydrates [33,34]. Therefore, impairing the growth and metabolism of these bacteria is desirable.

## 3. Silver Diamine Fluoride SDF: Sharpening an Anticaries Agent

As traditional approaches to the management of caries in young children are challenging, expensive, and often risky, there has recently been a marked shift towards minimally invasive approaches. SDF, as a topically applied antimicrobial agent, leverages the antimicrobial actions of several components: the alkaline pH (pH range from 9–13), the silver ions (25% by wt), fluoride ions (5% by wt), and ammonia (8% by wt). The balance of 62% is water, which acts as the solvent [35].

SDF was first introduced in Japan in the 1960s by Nishino, who added ammonia to improve on existing silver fluoride formulations [36]. While marketed with regulatory approval for treating dentinal hypersensitivity, SDF is used widely off-label for arresting dental caries in both deciduous and permanent teeth [37]. As a topical agent, it is more effective than fluoride varnishes [3]. SDF has demonstrated greater efficacy for halting the progress of lesions and gives increased fluoride absorption into tooth structure when compared with fluoride varnishes and topical fluoride gels, in both in vitro and in vivo studies [38,39].

The effectiveness of SDF in causing caries arrest is attributed to the combined actions of the high concentrations of fluoride and silver ions as well as the alkaline pH [40]. Ionic silver attacks bacterial microbes via multiple approaches, as explained in detail below. Moreover, the silver ions in SDF form a layer of silver phosphate that provides resistance to further decay, while fluoride ions convert hydroxyapatite to less soluble fluorapatite [41]. Together, these actions inhibit the further progression of demineralisation and help to preserve dentine collagen, protecting it from further degradation [42].

SDF has been produced commercially in various concentrations (12%, 30%, 38%, and 40%), with the 38% versions being in widespread use [4]. These formulations contain 44,800 ppm of fluoride, which is 8 times above the threshold needed for antimicrobial actions on bacteria [4]. When applied to primary teeth, 38% SDF has greater potency than 12% SDF for caries arrest [5,43]. Worldwide, multiple SDF brands are manufactured. Advantage Arrest^TM^ has a pH of 10 and contains 24.4–28.8% silver (*w/v*) and 5.0–5.9% fluoride. SDI Riva Star^TM^ has a pH of 13 and contains 35–40% (*w/v*) silver fluoride (AgF) and 15–20% (*w/v*) ammonia. The manufacturer also provides a solution of potassium iodide (KI) to be applied immediately onto the surface to scavenge precipitated silver [44]. The same manufacturer also has an ammonia-free neutral formulation (pH 7.4), SDI Riva Star Aqua™, with 40% AgF. Removing the ammonia was intended to prevent transient gingival/mucosal burns. Another notable product, CSDS, is made by Whiteley. This is also ammonia-free, and contains 40% AgF and 10% stannous fluoride (SnF_2_). The SnF_2_ acts as a reducer for excessive silver ions [45].

The use of silver fluoride in various forms can exert potent effects on microbial growth. A significant reduction in levels of *S. mutans* has been seen following the application of SDF onto an infected dentine surface in vitro [46]. A similar beneficial effect on impairing the growth of *Lactobacilli* [47] has also been seen in an in vitro study. The high concentration of fluoride ions (44,800 ppm) exerts a two-level action on bacteria. Firstly, it disrupts bacterial enzymes that regulate carbohydrate uptake and metabolism. Secondly, it impairs biofilm formation [47].

With regard to the interaction between SDF and tooth structure, there is some evidence that SDF remineralises dentine [38]. This occurs through the creation of silver phosphate and the deposition of calcium fluoride, both of which contribute to an increase in pH (from 5.5 to 9–13). The calcium fluoride, which is believed to remain on tooth surfaces via protein-based globules, can discharge fluoride ions in acidic mediums (cariogenic conditions) and serves as an effective medium-term fluoride reservoir [48]. The formation of fluorapatite with its reduced acid solubility lessens the impact of acids produced by the dental plaque biofilm [35]. Silver ions that react and bind with hydroxyapatite form a type of protection shield against future cariogenic attacks [42]. Additionally, hydrolytic collapse of dentine collagen is impaired via inhibition of proteolytic enzymes, including cathepsins (or cysteine cathepsins) and matrix metalloproteinases (MMPs) [49]. Thus, the effectiveness of SDF results from multiple pathways, a point that is very relevant when considering how to enhance the formulation or substitute other components for the silver.

The ionic silver in SDF creates several issues. Due to the alkaline pH of SDF, during application, the solution must be applied in very small amounts, and some manufacturers recommend that gingival protection be applied before SDF. Proper cotton roll isolation is important because of the risk of causing burns to skin or mucosa [50]. The fate of silver applied to deep lesions that are close to the dental pulp is a further point. In the 1990s, Gotjamanos et al. [51] reported the possibility of silver ions penetrating through the tooth structure and reaching the pulp chamber. This situation would occur when SDF was applied to very deep carious lesions that were close to the dental pulp [52]. A further point is that SDF has a noticeable, odd metallic taste that may cause momentary nausea [7].

Going beyond these short-term issues, the most significant drawback of using SDF, especially in high concentrations, is that it creates long-lasting black stains on the treated surfaces due to the formation of silver compounds, especially silver phosphate and silver sulphide [53]. Within two minutes of SDF application, the treated dentine surface darkens noticeably and irreversibly. This is followed by a gradual increase in the intensity of staining from 5 min to 5 h post-application. The maximum colour change following SDF application occurs at 12 h. The intensity of the stains varies according to the frequency of application [6]. Paradoxically, the discolouration has been considered an SDF success indicator, since it represents a zone high in calcium, phosphorus, fluoride, and silver ions [38].

While parents might accept SDF staining on primary posterior teeth, they have lower acceptance of its application to primary anterior teeth, as these are within the aesthetic zone [9]. They may accede to its use on anterior teeth as a fall-back measure when the child displays an uncooperative attitude during regular dental visits [54,55]. In contrast, parents of a cooperative child are more inclined to prefer aesthetic tooth-coloured restorations for the anterior teeth of their child in order to alleviate the sense of guilt from not having managed their child’s oral hygiene, and because reliable restorative techniques and procedures exist [56].

The black discolouration caused by SDF is an issue that could be addressed by using less silver or by using a scavenging agent. As mentioned previously, one manufacturer includes a topical application of saturated potassium iodide [57]. The rationale behind this is that the reaction between available silver ions and KI results in the precipitation of silver iodide (a bright yellow chemical compound) [58]. Silver iodide formation should minimise the concentration of free silver ions that eventually discolour the tooth surface. Despite this approach having been rated with “insufficient clinical evidence” in systematic reviews [59], in vitro studies have reported that the use of the SDF followed by KI application may lead to better aesthetic outcomes [60]. However, these outcomes appear to be temporary. A randomised clinical trial (RCT) showed no significant difference between the SDF/KI group and the SDF-only group at 30 months follow up [61]. They concluded that the further decomposition of silver iodide due to its photosensitivity leads to further release of silver and iodine. There is a concern that immediate application of KI following SDF may stain demineralised dentin [62]. This has prompted the search for a future antimicrobial agent that can achieve a balance between providing an adequate antibacterial effect and not significantly staining the tooth surface. Alternative potential antimicrobial agents include silver nanoparticles (AgNPs), selenium nanoparticles, and copper nanoparticles.

Nanoparticles (NPs) have diameters ranging from 1 to 100 nm. Their large surface area and strong chemical reactivity are desirable features [63]. Several metal-based nanomaterials are being actively considered for inclusion in dental materials and therapeutic products [14]. Furthermore, combining NPs with polymers and coating them with other nanocomposites provides an opportunity for multiple types of physio-chemical modification [64]. Each modified form can have unique chemical and antibacterial properties. A high level of antibacterial activity is an anticipated result of the strong interactions between certain metal NPs and the negatively charged surface of bacterial cells. These interactions occur because of the vast surface area and high charge density of NPs [65].

As the size of NPs reduces, their antibacterial properties improve. For instance, 10 nm AgNPs are small enough to penetrate the bacterial matrix and can cause an imbalance in vital cellular functions like DNA replication, particularly in Gram-negative bacteria [2]. Moreover, by triggering oxygen radical production, they cause lipid peroxidation, which disrupts bacterial cell membranes and decreases bacterial metabolism [66]. There have been some preliminary studies of AgNPs. Tirupathi and others assessed the anticariogenic capacity of AgNPs loaded into a sodium fluoride (NaF) varnish and compared this with 38% SDF [67]. It proved to be as effective as the SDF but without causing undesirable postoperative staining [67]. Additionally, the AgNPs did not generate silver oxide when exposed to oxygen in the medium; hence, the demineralised enamel did not stain black [68]. Furthermore, in an in vitro study, Targino and others compared 38% SDF and a nano form of silver fluoride, finding that AgNPs exert greater antimicrobial actions than silver ions [69]. This provides some support for the extension of this line of work. Likewise, the further development of hydroxyapatite NPs might result in the replacement of fluoride in topical anticaries agents, thusavoiding issues with fluoride in young patients [2].

## 4. Silver: The Old New

For thousands of years, the antimicrobial actions of silver have been recognised. In 335 BC, Alexander the Great used silver vessels for storing water and silver jugs for drinking [70]. The action of silver in keeping water from fouling was recognised before the advent of the “Germ Theory of Disease” [71]. Silver compounds were used in wound healing remedies by ancient Greeks due to their antimicrobial properties [72], and this idea now extends into the modern age to dressings for burns. Prior to the advent of antibiotics, silver was a widely used antibacterial agent [73].

In dentistry, silver nitrate (AgNO_3_) was used as a caries-arresting solution and an antibacterial agent since the 18th century [74]. In the 1840s, AgNO_3_ was used to treat caries in primary teeth and for control of gingival disease, marking the first documented use of silver compounds in dentistry. Thereafter, it was further used as a dentine desensitiser, cavity disinfectant, and caries preventive treatment for permanent molars [75,76,77]. In 1908, GV Black described using AgNO_3_ to treat caries in children as “the first measure against the disease” [78]. Then, in the 1960s, it was proposed that silver and fluoride should work together to prevent cavities; however, because of the discolouration of caries lesions, therapeutic use of silver fluoride compounds was limited [3].

Several modes of action of silver explain its antibacterial actions. Firstly, it attacks bacterial cell walls and changes their permeability, which leads to the leakage of cellular contents and impaired transport activity across the membrane [79]. Silver also generates reactive oxygen species (ROS) and hydroxyl radicals, which damage microbial DNA and cause lipid destruction through oxidation [80]. Ionic silver also inhibits the proton motive force, leading to apoptosis [81]. Moreover, AgNPs deactivate respiratory chain dehydrogenases, which inhibits cell respiration and growth. Once combined with halides, AgNPs release silver ions and produce more ROS [82]. A recent study by Hovhannisyan and colleagues pointed out that the effect of green synthesised AgNPs on bacterial membrane permeability was due to effects on formate hydrogenlyase proton–potassium transporters, energy-dependent H^+^-fluxes, and H^+^-translocating ATPase activity [83]. In addition, by specifically disturbing F_O_F_1_-ATPase activity, these biogenic AgNPs can affect bacteria that have no respiratory chain such as the *Enterococcaceae* species. The antimicrobial mechanisms of action of AgNPs are demonstrated in Figure 2.

## 5. SDF Re-Composition: Nanometals vs. Ionic Silver

Although the individual chemical ingredients of SDF work in cooperation to halt caries, the question arises as to whether the same or better clinical actions would be achieved with less staining by reformulation of the product. Hence, several research groups have evaluated the use of nano formulated approaches instead of traditional silver-based topical cariostatic agents [63,68,84,85,86]. Nano-silver fluoride is one of the suggested substitutes [63]. A study conducted by Nagireddy et al. used chemically synthesised AgNPs added to 0.05 ppm NaF and tested this formula for anticaries effects on 100 primary teeth. The results of their study showed 78% caries arrest within seven days. However, an issue with the design of their study was that normal saline was used as a control treatment, and there was no direct comparison to SDF [63].

In another study, SDF was modified by adding different concentrations of copper-doped bioglass nanoparticles (CuBGNPs), and the mixtures were assessed for their viscosity and antibacterial actions [84]. This modified form of SDF showed improved ion release and decreased cytotoxicity; in addition, a cumulative increase in the antimicrobial effect was observed as the concentration of CuBGNPs rose. Discolouration was not investigated.

The combination of fluoride ions and silver NPs has also been compared with SDF [86]; the two provide similar bactericidal effects, but the version with AgNPs does not cause noticeable tooth discolouration. Taken together, these results suggest that the use of nanoparticles of silver or of silver fluoride could be one way to solve the problem of discolouration.

## 6. Selenium: A Plausible Substitute

Selenium (Se) was identified as a natural element by the Swedish chemist Jon Jakob Berzeliusin 1817 [87]. It occurs naturally in limited amounts. Most commercial Se is refined from other metals such as copper (Cu) using electrolysis [88]. Se can be organic (selenocysteine and selenothionine) or inorganic (selenate and selenite). Based on its configuration, it can exist either in a crystal form (monoclinic and trigonal) or in an amorphous polymorphic form [89]. Se is one of the group-6 elements in the periodic table and is situated between non-metallic sulphur (S) and metallic tellurium (Te) in the oxygen group [90]. Thus, the chemistry of Se is somewhat similar to that of sulphur, although the two have different oxidation capacities [91]. Se shows multiple oxidation states: Se^2+^, Se^4+^, Se^6+^, and Se^2–18^.

### 6.1. Selenium and Human Health

Selenium is an essential micronutrient in the human diet. Generally, Se has a valuable impact on human health at different levels. It plays a vital role in several intracellular activities [92], and its sufficient intake is linked to protection against various diseases, including hypercholesterolemia, some malignancies, and cardiovascular illnesses [93]. Se has unique growth-regulating capabilities and fulfils a role as a cofactor for essential enzymes such as thioredoxin reductases (which control the redox state of the cell by inhibiting thioredoxin) and glutathione peroxidases (which prevent cellular destruction caused by oxygen-free radicals) [94]. Other health-related effects of Se include its antibacterial and antiviral actions [95]. As a trace element in the diet, Se is essential for human health [96]. Decreased Se levels have been correlated to impaired cognitive ability, weak immunity, and increased mortality risk [97].

### 6.2. Selenium Nanoparticle Properties: Size, Shape, and Synthesis

SeNPs have a core of inorganic Se (0). They can exert their own actions or can be loaded with additional therapeutic agents [98]. SeNPs have superior biosafety and biocompatibility to organic or inorganic Se molecules [99]. In addition, they exhibit distinctive chemical and physical characteristics owing to their large surface-to-volume ratio and high surface energy [100]. SeNPs have high bioavailability but low toxicity [101].

Additionally, the parameters of SeNPs can be adjusted to optimise their physical and chemical features, including through altering the synthesis process or by using capping agents [102]. SeNPs have been synthesised in a range of configurations, including spheres [103], cubes [104], ribbons [105], flowers [106], nanorods [107], and nanoneedles [108]. Generally, the most frequently used shape for therapeutic purposes is spherical [109]. Wire-shaped SeNPs have shown greater photoconductivity, while spherical-shaped SeNPs have demonstrated greater biocompatibility [110,111]. Furthermore, certain forms of SeNPs are generated by specific synthetic processes. For instance, green synthesis can result in SeNPs with a hexagonal ring form [112,113,114].

The adjustability of SeNPs extends to their charge. SeNPs are normally negatively charged, but this can be altered to positive by using surface modifiers such as chitosan [115,116]. This type of tuning is important in terms of binding to different types of microorganisms. In terms of the antimicrobial actions of SeNPs, one study reported that both Gram-positive and Gram-negative bacteria were inhibited to a similar extent by SeNPs [117]. In contrast, another study by Wang and others found that SeNPs were less effective against Gram-negative bacteria (*P. aeruginosa* and *E. coli)* after 72 h of treatment [118]. It is possible that SeNPs could be less active against Gram-negative bacteria due to electrostatic repulsion between negatively charged lipopolysaccharide in the bacterial cell membrane and SeNPs, unlike the situation with Gram-positive bacteria, which have a slight positive or neutral surface charge [119]. Although the antimicrobial impact of SeNPs has not yet been fully explored [120], this parameter has been demonstrated in several studies, and these are summarised in Table 1.

### 6.3. Selenium Nanoparticles and Toxicity Concerns

Although selenium is a trace element and is essential for human body, there is a fine line between healthy therapeutic levels of selenium and a harmful toxic concentration. As a supplement, the World Health Organisation recommends 55 µg of selenium per day for healthy individuals [121]. Selenium compounds, especially at high concentrations, exhibit noticeable toxicity. On the contrary, SeNPs have better safety use and cause less cytotoxicity [110]. In vivo evidence has shown that the SeNP lethal dose is 4–6 times less than other formulations of organic and inorganic selenium [122]. In general, the cytotoxic actions of SeNPs are closely associated with their ability to damage cell membranes and to generate oxidative stresses [123]. Furthermore, stabilisation of SeNPs could play a part in further reducing their toxicity [107]. For example, polysaccharide-stabilised SeNPs show less toxicity and enhanced bioavailability [110].

It has been reported that if human selenium consumption surpasses 400 µg/day, symptoms of selenosis (selenium intoxication) might be observed. Selenosis causes symptoms that range from hair and nail damage to severe neuromotor disturbances. However, other studies argued that a much higher intake (e.g., 3000 µg/day) is needed to cause symptoms of selenosis [123]. The median lethal dose of SeNPs is 92.1 mg/kg, while it is only 14.6 mg/kg for seleno-methylselenocysteine [101]. The cytotoxicity of SeNPs varied according to their synthesis method. Biogenic (green) synthesis results in less toxic forms of nano selenium compared with chemically synthesised SeNPs [124]. Lastly, in terms of genotoxic impacts of SeNPs, some studies have found that bare (uncoated) SeNPs cause a dose-dependent genotoxic action; however, this problem can be reduced markedly by capping or coating SeNPs with surfactants [107]. Overall, in terms of clinical use under controlled conditions where little or no ingestion is likely to occur, concerns over the cytotoxic effects of SeNPs are likely to be low. Moreover, this concern can be minimised by using green synthesis and by applying suitable capping agents.

### 6.4. SeNPs and the Role of Capping Agents

Stabilisers/capping agents can be added to chemically synthesised SeNPs to enhance their physio-chemical characteristics and maintain their bioactivity and size by enhancing their stability and preventing them from agglomerating [125]. Furthermore, such stabilisation influences the ultimate size of SeNPs and, thus, is strongly related to antioxidant capacity, cytotoxicity, and antimicrobial actions [93]. Multiple agents have been used to coat SeNPs. Chitosan is a type of chitin that can be extracted from insects and crustaceans. Chitosan-capped SeNPs have reasonable bioavailability, a positive charge, and minimal toxicity; thus, they have been used in several therapeutic applications [126]. Various polymers have also been used to stabilise SeNPs, including polyvinyl alcohol, polyethylenimine, and polyvinylpyrrolidone [110]. Plant-derived polysaccharides have been used to cap SeNPs, and this enhances their antioxidant actions [127,128]. Proper selection of capping agents can be used to optimise the chemical, physical, and biological properties of selenium nanoparticles.

**Table 1 antibiotics-12-00298-t001:** Summary of published research on antimicrobial action of selenium nanoparticles.

STUDY	METHODS	KEY FINDINGS
[103]	Synthesis: chemically—Na_2_SeO_3_ reduction; Stabiliser: BSA Characterisation: TEM and DLS Tested microbes: *S. aureus* SeNPs tested concentrations: 7.8, 15.5, 31 μg/mL	-SeNPs shape/size: spherical/40–60 nm -SeNPs inhibited *S. aureus* growth compared with no treatment. -SeNPs killed approx. 40% of *S. aureus* after 3, 4, and 5 h
[129]	Synthesis: chemically—chitosan dissolution Characterisation: DLS Sample: skin infection swabs (*n* = 25) Tested microbes: 49 various bacterial strains Antibacterial tests: agar diffusion assay, growth curves SeNPs tested concentrations: up to 100 μg/mL	* Size and shape of SeNPs: not revealed -Bacterial growth curves were inhibited by low concentration (1 μg/mL) in all tested bacterial isolates -64 μg/mL of SeNPs shows complete inhibition when applied on *E. fergusonii*, *P. aeruginosa*, and *S. agalactiae* assays
[130]	Synthesis: biosynthesis—R. *Alstonia eutropha* bacterium-based Na_2_SeO_3_ reduction Characterisation: TEM, SEM, XRD, and SAED Tested microbes: *E. coli*, *P. aeruginosa*, *S. aureus*, *S. pyogenes*, and *A. clavatus*	-SeNPs shape/size: spherical 40–120 nm hexagonal crystalline -MIC: *E. coli* 125 μg/mL, *P. aeruginosa* 100 μg/mL, *S. aureus* 100 μg/mL, *S. pyogenes* 250 μg/mL and *A. clavatus* 500 μg/mL
[131]	Synthesis: chemically—Na_2_SeO_3_ reduction using glutathione Stabiliser: BSA Characterisation: FTIR, UV–Vis, DLS, SEM, and TEM Tested microbes: *E. coli*, *S. aureus*, *Salmonella*, and *Listeria*	-SeNPs shape/size: spherical ~79 nm -SeNPs show dose-dependent antimicrobial action against *S. aureus* but not the other three pathogens -SeNPs exerts cytotoxicity on cancer cells Caco-2 after 24 h exposure
[132]	Synthesis: (1) biosynthesised SeNPs (*B. mycoides*); (2) chemically synthesised SeNPs Characterisation: TEM, EDX, and DLS Tested microbes: *P. aeruginosa* and *S. aureus* Antibacterial tests: biofilms on hydroxyapatite discs	-SeNPs shape: spherical -Biosynthesized SeNPs had greater antibacterial action than chemically synthesized SeNPs
[133]	Synthesis: chemical method (using SeO_2_ as a precursor) Stabiliser: polyvinyl alcohol Characterisation: TEM, EDS, and FTIR Tested microbes: *S. aureus* Cytotoxicity: human dermal fibroblasts	-SeNPs shape/size: amorphous and spherical 43–205 nm -SeNPs antimicrobial; action is size dependent -greatest antibacterial action was observed in 81 nm SeNPs
[134]	Synthesis: chemical method (using SeO_2_ and ascorbic acid) Stabiliser: egg white lysozyme (EWL) Characterisation: UV–Vis, FTIR, TEM, and XRD Tested microbes: *E. coli*, *S. pneumoniae*, *B. cereus*, *K. pneumoniae*, *P. mirabilis*, and *B. subtilis*	-SeNPs shape/size: spherical (crystalline structure)/40–60 nm -MIC: 10.0 μg mL^−1^ -ZOI: 19 mm (*B. subtilis)*, 15 mm *(E. coli)*, 14 mm *(B. cereus)*, and 13 mm *(K. pneumoniae)* -SeNPs exhibits prolonged stability (minimum of 12 months)
[135]	Synthesis: chemical method (using dissolution of Na_2_SeO_3_, L-ascorbic acid, and polyvinyl alcohol PVA in purified water) Characterisation: TEM, XPS, and SEM (Ti implants coated with SeNPs 30–70 nm) Tested microbes: *S. aureus*	-SeNPs shape/size: spherical (crystalline structure)/50–200 nm -Oxidation state: zero -SeNPs showed great antibacterial activity even at 0.5 ppm
[136]	Synthesis: chemical method (using Na_2_SeO_3_ as a precursor) Stabiliser: BSA, D-glucose, and soluble starch Characterisation: UV–Vis, FTIR, SEM, and EDXTested microbes: *B. subtilis* (mid-log) Antimicrobial test: SEM	-SeNPs shape/size: (BSA): rod shaped/200–250 nm (D-Glucose): spherical/200 nm (starch): cubes/250–300 nm -Antioxidation capacity: starch > D-Glucose > BSA -Antibacterial effect seen in all samples but not quantified
[137]	Synthesis: green synthesis—ascorbic acid as reductant Stabiliser: BSA Tested microbes: *S. aureus* and *E. coli*	-SeNPs shape/size: spherical/10–100 nm Proliferation of fibroblasts was promoted by SeNPs, whereas growth of *S. aureus* was suppressed
[138]	Synthesis: green method (Na_2_SeO_3_ as a precursor, using bovine urine) Characterisation: UV–Vis, SEM, TEM, DLS, AFM, and EDX Tested microbes: *E. coli*, *K. pneumoniae*, *P. aeruginosa*, *Serratia*, *Proteus*, and *S. aureus*	-SeNPs shape/size: spherical/110 nm -SeNPs effective against all species especially against *Klebsiella* species.
[139]	Synthesis: chemically—using Na_2_SeO_3_ Characterisation: UV–Vis Tested microbes: *P. aeruginosa*, *Salmonella typhimurium*, *E.coli*, *S. sanguinis*, *S. aureus*, and *E. faecalis*	MIC: *S. sanguinis*, 68 μg/mL *S. aureus*, 137 μg/mL *E. faecalis* 274 μg/mL
[140]	Synthesis: chemically—Na_2_SeO_3_ as a precursor Stabiliser: (i) BSA + ascorbic acid, (ii) Chitosan + ascorbic acid, and (iii) glucose Characterisation: FTIR, XRD, DLS, TEM, DLS, and UV–Vis Tested microbes: mixed biofilm *S. aureus* and *C. albicans* *Cytotoxicity*: human dermal fibroblasts	-SeNPs shape/size: spherical/70–300 nm -MIC: The lowest MIC was against *C. albicans* (25 μg/mL) -Zeta potential: SeNPs-Chit < SeNPs-BSA < SeNPs-Gluc * SeNPs-BSA was less cytotoxic than the other two formulations
[141]	Synthesis: chemical method (using Na_2_SeO_3_ as a precursor) Characterisation: UV–Vis, XRD, and SEM Tested microbes: *S. aureus* isolated from public water Antibacterial tests: disk diffusion assay, microdilution assay	-SeNPs shape/size: rod/85–275 nm -MIC: 50 μg/mL -synthetic SeNPs contributed to the breakdown of *S. aureus* biofilms
[109]	Synthesis: chemically—reduction of Na_2_SeO_3_ Characterisation: TEM, DLS, and XRD Tested microbes: *E. faecalis* *Cytotoxicity*: human fibroblasts	-SeNPs shape/size: spherical/77 + 27 nm -CFU: significantly reduced compared with control -Methylene Blue-induced Photodynamic Antimicrobial Chemotherapy SeNPs-MB-PACT showed the greatest antibiofilm impact -At 128 μg/mL SeNPs, 50% of human fibroblasts survived
[142]	Synthesis: biosynthesis—guava leaves (*Psidium guajava*) Characterisation: UV–Vis, DLS, TEM, and XRD Tested microbes: *E. faecalis* Group I: Distilled water (control), Group II: SeNPs (1 mg/mL), Group III: Calcium hydroxide (1 mg/mL), Group IV: 2% CHX, and Group V: 5.25% NaOCl Antibacterial tests: agar diffusion method, microdilution, viable cell count, antibiofilm assay, and Anthrone and Bradford’s tests	-SeNPs shape/size: spherical/30–50 nm -Zeta potential: −60 mV. -MIC: 25 mg/mL -ZOI: SeNPs 11.33–28.50 mm (based on the concentration) -SeNPs are the most active against *E. faecalis* biofilm, followed by NaOCl, CHX and Ca(OH)_2_
[19]	Synthesis: green method (using *Brassica Oleracea* (Broccoli)) Characterisation: UV–Vis, TEM, FTIR, and EDX Tested microbes: *S. mutans*, *S. aureus*, *E. faecalis*, *Lactobacillus*, and *C. albicans*	-SeNPs shape/size: spherical/10–25 nm -greatest ZOI reported against *S. mutans*
[143]	Synthesis: biosynthesis—Na_2_SeO_3_ with citrus fruit extracts (lemons and grapefruits) Characterisation: UV–Vis, TEM, FTIR, and DLS Tested microbes: *E. coli*, *M. luteus*, *K. pneumoniae*, and *B. subtilis* Antimicrobial tests: agar diffusion assay	-SeNPs shape/size: not determined -SeNPs ZOI: 20 ± 1.646 mm against *K. pneumoniae* -A noticeable antimicrobial effect was detected after testing Citrus SeNPs -SeNPs could replace traditional antibiotics
[144]	Synthesis: biosynthesis—Na_2_SeO_3_ using *Rosmarinus officinalis* extract Characterisation: UV–Vis, DLS, TEM, XRD, and FTIR Tested microbes: *M. tuberculosis*, *S. aureus*, *S. mutans*, *E. coli*, and *P. aeruginosa*	-SeNPs shape/size: Spherical 20 to 40 nm -MIC in µg/mL: *(M. tuberculosis)*, 256; *(S. aureus*), 16; *(S. mutans)*, 32; (*E. coli*), 128; and *(P. aeruginosa)*, 64
[145]	Synthesis: chemical reduction Bacterial tests: MHB microdilution to assess MIC and MBC	SeNPs size: 81.4 nm MIC in µg/mL: *S. mutans* 68, *L. acidophilus* 137 and *C. albicans* 274 MBC in µg/mL: *S. mutans* 274 after 1–2 h and 137 after 6–24 h
[120]	Synthesis: biosynthesis—Na_2_SeO_3_ (precursor) using *Calendula officinalis* L. flowers as a (capping agent) Characterisation: SEM, TEM, FTIR, and EDX Tested microbes: *Serratia marcescens*, *Enterobacter cloacae*, and *Alcaligenes faecalis bacteria* Antimicrobial tests: disc diffusion	-SeNPs shape/size: spherical/40–60 nm -SeNPs demonstrated superior antibacterial effect, at various incubation times compared with the antibiotic ciprofloxacin CIP -SeNPs had greater antioxidant activity than methanolic extracts of flowers Cof-Met extract and Na_2_SeO_3_
[146]	Synthesis: chemically—concentration was 128 and 64 μg/mL Characterisation: chemical reduction method *Cytotoxicity*: MTT test on human gingival fibroblast Tested Microbes: *S. mutans*	SeNPs enhanced photodynamic therapy (PDT) activity and exhibited significant antibiofilm action against *S. mutans*
[15]	Tested Microbes: *S. salivarius*, *S. mutans*, and *S. sanguinis* *Control*: untreated sealant *Analysis*: SEM, CLSM	Organo-selenium dental sealant was able to inhibit the growth of all species individually and in a mixed biofilm.

Abbreviations: AFM: Atomic force microscopy; BSA: Bovine serum albumin; CHX: Chlorhexidine gluconate; CLSM: Confocal laser scanning microscopy; DLS: Dynamic light scattering; EDX: Energy dispersive X-ray analysis; EWL: Egg white lysozyme; FTIR: Fourier transform infrared; Na_2_SeO_3_: Sodium selenite; MBC: Minimum bactericidal concentration; MB-PACT: Methylene Blue-induced photodynamic antimicrobial chemotherapy; MHB: Mueller Hinton broth; MIC: Minimum inhibitory concentration; MTT: 3-(4,5-dimethylthiazol-2-yl)-2,5-diphenyl-2H-tetrazolium bromide assay; NaOCl: Sodium hypochlorite; SAED: Selected area (electron) diffraction; SEM: Scanning electron microscope; SeO_2_: Selenium dioxide; TEM: Transmission electron microscope; UV–Vis: Ultraviolet–visible; XPS: X-ray photoelectron spectroscopy; XRD: X-ray diffraction analysis. ZOI: Zone of inhibition.

As shown in Table 1, SeNPs possess microbial inhibition action against a variety of bacterial strains as well as against important fungi (*C. albicans*), and this action varies according to the synthesis method and the SeNP properties. Some studies have adopted eco-friendly green synthesis using plants, fruit, vegetables, and natural substances. Others have followed traditional chemical synthesis approaches and then used capping agents. Most studies of SeNPs began with Na_2_SeO_3_ as the precursor molecule. A noteworthy pattern across these studies was that, regardless of the variables being examined, all the microbes used were affected by exposure to SeNPs to some extent. A limitation of the work to date is that most studies used organisms in the planktonic state rather than in biofilms; moreover, biofilm studies were single species rather than multispecies. Hence, additional work is needed to determine the optimum profile of SeNPs for achieving caries-arresting actions.

## 7. Conclusions and Future Directions

SDF has become a popular treatment choice for halting dental caries in young children, especially children with minimal cooperation for dental procedures. However, it darkens hard tooth structure permanently, which compromises aesthetics, particularly in the anterior smile zone. This review identifies several methods that could be followed to reduce this concern, including the use of AgNPs and NPs of silver fluoride. There could also be value in using remineralising agents other than fluoride, such as NPs of hydroxyapatite. There could be variations made to formulations in order to lower the levels of silver and fluoride in the SDF or even to replace one or both of the silver and fluoride components completely. Moreover, since oxidation processes appear central to the chemistry of the staining, adding SeNPs which have antioxidant actions could have an anti-staining benefit; SeNPs could be used for their antimicrobial actions as well. Future research should address the topic of selenium chemistry to optimise how SeNPs could be used with or in place of ionic silver. Incorporating other antimicrobial metals as nanoparticles such as copper, zinc oxide, and cerium oxide should also be deeply explored in the context of topical anticaries agents, taking into account the optimal physicochemical parameters for each of these.

## Figures and Tables

**Figure 1 antibiotics-12-00298-f001:**
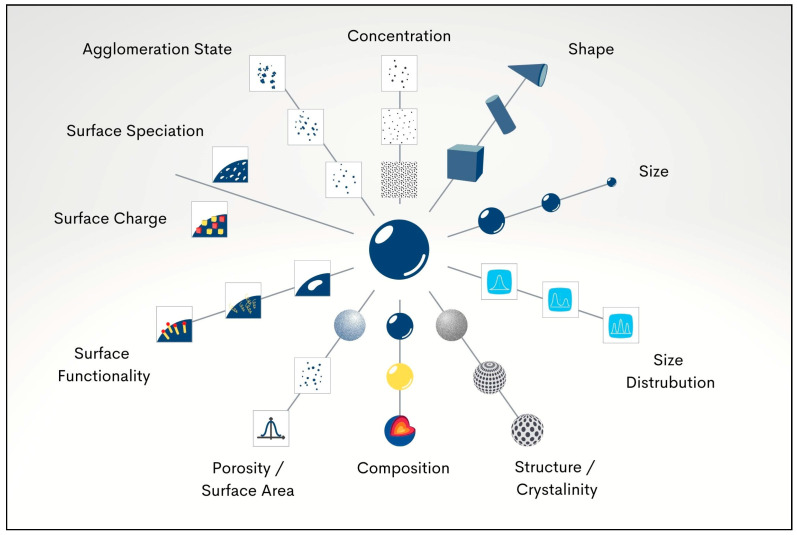
Physio-chemical parameters which determine the biological impact of nanoparticles.

**Figure 2 antibiotics-12-00298-f002:**
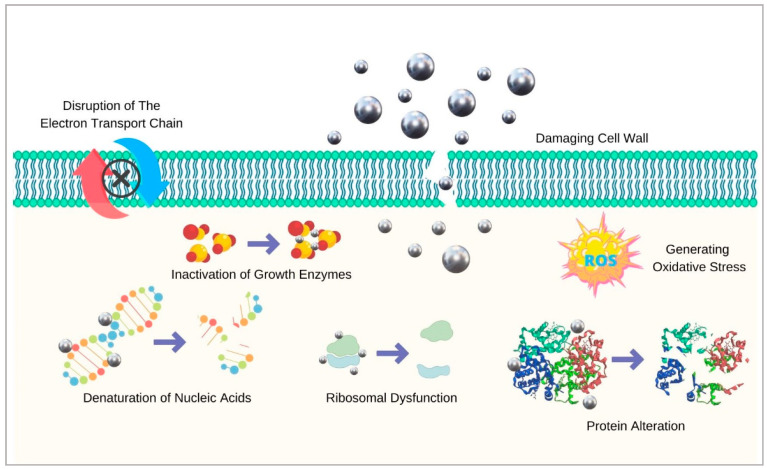
Antimicrobial mechanisms of silver nanoparticles.

## Data Availability

Not applicable.

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
