# Peer review of "Amelioration Strategies for Silver Diamine Fluoride: Moving from Black to White"

_antibiotics, 2023, doi:10.3390/antibiotics12020298_

Round 1
Reviewer 1 Report
The authors make a complete review of the strategies to improve the adverse effects of Silver diamine fluoride (SDF), this review seems important to me. However, I suggest not using so many abbreviations that are not really necessary because there is an excess in the whole document, which makes it tedious to read.
Figure 1 is not very clear to me; I think it does not give enough information to include it in the review
It is not appropriate to synthesize the names of the bacteria as Streptococcus mutans (SM) is doing, as a taxonomic rule the first time it appears in the text it must be written completely and in italics, as you show, the following times the initial of the genus in uppercase period and then the complete species in lowercase and italics Eg: S. mutans
I ask you to please modify all the names of bacteria in this way.
In table 1. I do not see the title of the table and I think there is an excess of abbreviations, I suggest reviewing and putting the ones that are really necessary and leaving the full names better, as well as the paragraph “Microbes: A. clavatus: Aspergillus clavatus; B. cereus: Bacillus cereus; B. mycoides: Bacillus mycoides; B. subtilis: Bacillus subtilis; C. albicans: Candida albicans; E. coli: Escherichia coli; E. faecalis: Enterococcus faecalis; E. fergusonii: Escherichia fergusonii; K. pneumoniae : Klebsiella pneumoniae; M. luteus: Micrococcus luteus; M. tuberculosis: Mycobacterium tuberculosis; P. aeruginosa: Pseudomonas aeruginosa; P. mirabilis: Proteus mirabilis; S. agalactiae: Streptococcus agalactiae; S. aureus: Staphylococcus aureus; S. pneumoniae: Streptococcus pneumoniae; S. pyogenes: Strepto-coccus pyogenes; S. salivarius Streptococcus salivarius; S. sanguinis: Streptococcus sanguinis”.
The names of the bacteria should be put in with the taxonomic rules of writing so this should be omitted.
Author Response
The authors have re-organised the manuscript as suggested by the reviewer, and all the changes in the revised manuscript are highlighted in yellow
The authors have re-organised the manuscript as suggested by the reviewer, and all the changes in the revised manuscript are highlighted in yellow
Responses to Reviewer 1 Comments
Comment 1: The authors make a complete review of the strategies to improve the adverse effects of Silver diamine fluoride (SDF), this review seems important to me. However, I suggest not using so many abbreviations that are not really necessary because there is an excess in the whole document, which makes it tedious to read.
Response: We have amended this section as per reviewer’s suggestion.
Comment 2: Figure 1 is not very clear to me; I think it does not give enough information to include it in the review
Response: We have amended the Figure and provided further explanation.
Comment 3: It is not appropriate to synthesize the names of the bacteria as Streptococcus mutans (SM) is doing, as a taxonomic rule the first time it appears in the text it must be written completely and in italics, as you show, the following times the initial of the genus in uppercase period and then the complete species in lowercase and italics Eg: S. mutans. I ask you to please modify all the names of bacteria in this way.
Response: We have updated the names of the bacteria throughout the manuscript.
Comment 4: In table 1. I do not see the title of the table and I think there is an excess of abbreviations, I suggest reviewing and putting the ones that are really necessary and leaving the full names better, as well as the paragraph “Microbes: A. clavatus: Aspergillus clavatus; B. cereus: Bacillus cereus; B. mycoides: Bacillus mycoides; B. subtilis: Bacillus subtilis; C. albicans: Candida albicans; E. coli: Escherichia coli; E. faecalis: Enterococcus faecalis; E. fergusonii: Escherichia fergusonii; K. pneumoniae : Klebsiella pneumoniae; M. luteus: Micrococcus luteus; M. tuberculosis: Mycobacterium tuberculosis; P. aeruginosa: Pseudomonas aeruginosa; P. mirabilis: Proteus mirabilis; S. agalactiae: Streptococcus agalactiae; S. aureus: Staphylococcus aureus; S. pneumoniae: Streptococcus pneumoniae; S. pyogenes: Strepto-coccus pyogenes; S. salivarius Streptococcus salivarius; S. sanguinis: Streptococcus sanguinis”.
Response: The title of the table has been added on the top of it. Names of microbes have been removed as suggested
Comment 5: The names of the bacteria should be put in with the taxonomic rules of writing so this should be omitted.
Response: names of microbes were reviewed based on the taxonomic “capital letter for the genus name and an epithet beginning by a lowercase letter for the species name”
Reviewer 2 Report
Article titled “Amelioration Strategies for Silver Diamine Fluoride: Moving from Black
to White” is well-designed and touch very important and urgent problem of oral cavity infection diseases.
However, there are some deficiencies that need to be corrected, namely:
1. There was not any information concerning the:
A) shelf life of SeNPs, toxicity (cytotoxicity, genotoxicity),
B) the possibility of cumulative effect on organism,
C) the application peculiarities of appropriate stabilizing and capping agents,
D) the differences between the effect on Gram-positive and Gram-negative microorganisms.
2. There were some minor technical errors.
3. On the page 2015 (Figure 2) there was information on possible mechanisms of influence of nanoparticles on bacterial cells. Please explain also some other mechanisms of AgNPs action. For instance, Hovhannisyan et al., (2022, https://doi.org/10.3390/antibiotics11101415) mention also the influence of these particles on formate hydrogen lyase (FHL) proton-potassium transporter and, besides ETC disruption it influence also on F0F1-ATPase activity. Thus these nanoparticles can also influence on bacteria with the lack of respiratory chain (for instance, on bacteria from Enterococcaceae family).
Author Response
RESPONSES TO COMMENTS
All the changes in the revised manuscript are highlighted in yellow.
Responses to Reviewer 2 Comments
Comment 1: Article titled “Amelioration Strategies for Silver Diamine Fluoride: Moving from Black to White” is well-designed and touch very important and urgent problem of oral cavity infection diseases. However, there are some deficiencies that need to be corrected, namely:
- There was not any information concerning the:
- shelf life of SeNPs, toxicity (cytotoxicity, genotoxicity),
- the possibility of cumulative effect on organism,
- the application peculiarities of appropriate stabilizing and capping agents,
- the differences between the effect on Gram-positive and Gram-negative microorganisms.
Response: thank you for your valuable revisions. They are extremely important to be addressed in the review. Here are the responses to each point:
- A separate section has been added (lines 283-300).
- There is limited research on the cumulative effect of selenium nanoparticles from a therapeutic point of view. Though this drag the attention to a gap in the literature that must be explored.
- A split section has been added (Lines 303-313).
- A part has been added (lines 275-280).
Comment 2. There were some minor technical errors. On the page 2015 (Figure 2) there was information on possible mechanisms of influence of nanoparticles on bacterial cells. Please explain also some other mechanisms of AgNPs action. For instance, Hovhannisyan et al., (2022, https://doi.org/10.3390/antibiotics11101415) mention also the influence of these particles on formate hydrogen lyase (FHL) proton-potassium transporter and, besides ETC disruption it influence also on F0F1-ATPase activity. Thus these nanoparticles can also influence on bacteria with the lack of respiratory chain (for instance, on bacteria from Enterococcaceae family).
Response: This section is edited as suggested by the reviewer (220-224)
Reviewer 3 Report
This paper “Amelioration Strategies for Silver Diamine Fluoride: Moving from Black to White” is about the effect of SDF. Basically, the key points are related to how to reduce staining and the possibility of using alternative nanomaterial-based antimicrobial agents.
.
The paper structure and the overall content are good.
Nevertheless, I suggest some improvements be performed before this manuscript can be considered suitable for publication.
Lines 28-9 AND/OR lines 76-86: The authors can also underline the role that the pandemic has had in the worsening of clinical scenarios such as ECC.
The authors could add a sentence like:
“The recent pandemic situation has further aggravated the situation as parents have inevitably delayed their children's visits to the dentist”. The authors coul cite: Paolone G, Mazzitelli C, Formiga S, Kaitsas F, Breschi L, Mazzoni A, Tete G, Polizzi E, Gherlone E, Cantatore G. One-year impact of COVID-19 pandemic on Italian dental professionals: a cross-sectional survey. Minerva Dent Oral Sci. 2022 Aug;71(4):212-222. doi: https://doi.org/10.23736/S2724-6329.21.04632-5
Lines 49-51 ALSO for lines 160-2: In a systematic review, parents’ decision whether to accept or reject SDF treatment was linked to tooth position; and a low acceptance rate was reported if SDF treatment was proposed for anterior teeth.
The reviewer thinks that the reader could benefit from knowing possible alternatives for the treatment of ECC in anterior teeth for the (few) cooperative patients.
The authors could add a sentence like:
“Parents are more inclined to prefer aesthetic restorations for the front teeth both to alleviate the sense of guilt of not having been able to manage their children's oral hygiene and due to the fact that reliable restorative techniques and procedures exist.” The authors could cite https://doi.org/10.3390/sym13050797 to support this sentence.
Line 62: add a space after the full stop.
Lines 147-8
The readers could wonder which type of isolation is necessary: cotton rolls? Liquid rubber dam? Absolute isolation with rubber dam?
Figure 1 and 2: are these pictures made by the authors, or are taken from other papers? If they are taken by other papers, they should have the correct license (CC) or permission shall be asked to the copyright holder.
Author Response
All the changes in the revised manuscript are highlighted in yellow.
Responses to Reviewer 3 Comments
This paper “Amelioration Strategies for Silver Diamine Fluoride: Moving from Black to White” is about the effect of SDF. Basically, the key points are related to how to reduce staining and the possibility of using alternative nanomaterial-based antimicrobial agents. The paper structure and the overall content are good. Nevertheless, I suggest some improvements be performed before this manuscript can be considered suitable for publication.
Comment 1: Lines 28-9 AND/OR lines 76-86: The authors can also underline the role that the pandemic has had in the worsening of clinical scenarios such as ECC.
The authors could add a sentence like: “The recent pandemic situation has further aggravated the situation as parents have inevitably delayed their children's visits to the dentist”. The authors coul cite: Paolone G, Mazzitelli C, Formiga S, Kaitsas F, Breschi L, Mazzoni A, Tete G, Polizzi E, Gherlone E, Cantatore G. One-year impact of COVID-19 pandemic on Italian dental professionals: a cross-sectional survey. Minerva Dent Oral Sci. 2022 Aug;71(4):212-222.
Response: We appreciate your comment, and have added such as sentence in the manuscript (lines 84-85).
Comment 2. Lines 49-51 ALSO for lines 160-2: In a systematic review, parents’ decision whether to accept or reject SDF treatment was linked to tooth position; and a low acceptance rate was reported if SDF treatment was proposed for anterior teeth. The reviewer thinks that the reader could benefit from knowing possible alternatives for the treatment of ECC in anterior teeth for the (few) cooperative patients.
The authors could add a sentence like: “Parents are more inclined to prefer aesthetic restorations for the front teeth both to alleviate the sense of guilt of not having been able to manage their children's oral hygiene and due to the fact that reliable restorative techniques and procedures exist.” The authors could cite https://doi.org/10.3390/sym13050797 to support this sentence.
Response: A similar sentence has been added as suggested (lines 167-170).
Comment 3. Line 62: add a space after the full stop.
Response: The space has been added as suggested
Comment 4. Lines 147-8. The readers could wonder which type of isolation is necessary: cotton rolls? Liquid rubber dam? Absolute isolation with rubber dam?
Response: The information point “cotton roll isolation” has been added to the sentence (line 153).
Comment 5. Figure 1 and 2: are these pictures made by the authors, or are taken from other papers? If they are taken by other papers, they should have the correct license (CC) or permission shall be asked to the copyright holder.
Response: Both figures were created entirely by ourselves as the authors so there is no need to seek permissions.